# Stability of Random Forests and Coverage of Random-Forest Prediction Intervals

**Yan Wang**
Department of Mathematics
Wayne State University
Detroit, MI 48202
`wangyan@wayne.edu`

**Huaiqing Wu, Dan Nettleton**
Department of Statistics
Iowa State University
Ames, IA 50011
`{isuhwu,dnett}@iastate.edu`

## Abstract

We establish stability of random forests under the mild condition that the squared response $(Y^2)$ does not have a heavy tail. In particular, our analysis holds for the practical version of random forests that is implemented in popular packages like `randomForest` in R. Empirical results show that stability may persist even beyond our assumption and hold for heavy-tailed $Y^2$. Using the stability property, we prove a non-asymptotic lower bound for the coverage probability of prediction intervals constructed from the out-of-bag error of random forests. With another mild condition that is typically satisfied when $Y$ is continuous, we also establish a complementary upper bound, which can be similarly established for the jack-knife prediction interval constructed from an arbitrary stable algorithm. We also discuss the asymptotic coverage probability under assumptions weaker than those considered in previous literature. Our work implies that random forests, with its stability property, is an effective machine learning method that can provide not only satisfactory point prediction but also justified interval prediction at almost no extra computational cost.

## 1 Introduction

Random forests (RFs) is a successful machine learning method that serves as a standard approach to tabular data analysis and has good predictive performance [10, 7]. However, there is a big gap between the empirical effectiveness of RFs and the limited understanding of its properties. Most known theoretical results are established for variants of RFs not necessarily used in practice [5, 25, 22, 14, 30]. For the RF version implemented in packages like `randomForest` in R [21], little is known without strong assumptions [6, 26, 35]; RFs is notoriously difficult to analyze as a greedy algorithm. Here we show an important property for the RF used in practice (as well as for other variants) under realistic conditions.

### 1.1 Stability of random forests

The first main contribution of this work establishes the stability condition for the RF.

**Theorem 1** (Stability of random forests, informal)**.** *For independent and identically distributed (iid) training data points $(X_i, Y_i), i \in \{1, \ldots, n\} \equiv [n]$ and a test point $(X, Y)$, if the squared response $Y^2$ does not have a heavy tail, then the RF predictor $\mathsf{RF}_B$ and any out-of-bag (OOB) predictor $\mathsf{RF}_B^{\backslash i}$ predict similar values, i.e.,*

$$\mathbb{P}\left(\left|\mathsf{RF}_B(X) - \mathsf{RF}_B^{\backslash i}(X)\right| > \varepsilon_{n,B}\right) \leqslant \nu_{n,B}, \tag{1}$$

37th Conference on Neural Information Processing Systems (NeurIPS 2023).

*where* $\mathsf{RF}_B$ *results from the aggregation of all $B$ base tree predictors, while* $\mathsf{RF}_B^{\backslash i}$ *only those with the point $(X_i, Y_i)$ excluded in training; $\varepsilon_{n,B}$ and $\nu_{n,B}$ are small numbers depending on $n$ and $B$.*

This result is referred to as the stability of the RF because it indicates that no single training point is extremely important in determining $\mathsf{RF}_B$ in a probabilistic sense. Theorem 1 relies on a recent important work that establishes the absolute stability (see below for a precise definition) of general bagged algorithms with bounded outputs [28]. We take advantage of the fact that the range of the RF output is conditionally dependent upon the maximal and minimal values of $Y$ in the training set, and then we show in theory that the stability property of the RF is possible even if $Y$ is marginally unbounded. To our knowledge, this is the first stability result established for the RF.

The technique used in our analysis requires that $Y^2$ not have a heavy tail (to make $\varepsilon_{n,B}$ and $\nu_{n,B}$ small). Though arguably already mild, we conjecture that this condition might be further relaxed. As shown below, numerical evidence suggests that the light-tail assumption may not be necessary for RF stability, which could hold even when $Y$ follows a heavy-tail distribution like the Cauchy distribution.

## 1.2  Random-forest prediction intervals

Stability is a crucial property of a learning algorithm. For example, stability has a deep connection with the generalization error of an algorithm [9, 19, 23]. Moreover, stability also turns out to be important in distribution-free predictive inference. In particular, an algorithm being stable justifies the jackknife prediction interval (PI), which otherwise has no coverage guarantee [3].

In this work, we show that stability makes it possible to construct a PI with guaranteed coverage from the OOB error of the RF. The OOB error is defined as $R_i = |Y_i - \mathsf{RF}_B^{\backslash i}(X_i)|$, $i \in [n]$. A main reason why such a PI is appealing is that $R_i$ can be obtained almost without extra effort. For example, a one-shot training using the R package `randomForest` gives us an RF predictor $\mathsf{RF}_B$ and all $n$ OOB predictions $\mathsf{RF}_B^{\backslash i}(X_i)$. So, from the computational point of view, a convenient way to construct a PI for a test point $(X, Y)$ is of the form "$\mathsf{RF}_B(X) \pm$ proper quantile of $\{R_i\}$" [17, 35].

The second main contribution of this work constructs such PIs and theoretically proves, under mild conditions, the non-asymptotic lower and upper bounds for the coverage probability.

**Theorem 2** (Coverage lower bound, informal). *Under the same assumptions as in Theorem 1, and for $\alpha \in (0, 1)$ [1], we have the following lower bound of coverage probability:*

$$\mathbb{P}\left(|Y - \mathsf{RF}_B(X)| \leqslant \text{the } \lceil (n+1)(1-\alpha) \rceil \text{-th smallest } R_i + \varepsilon_{n,B}\right) \geqslant 1 - \alpha - O(\sqrt{\nu_{n,B}}),$$

*where $\lceil \cdot \rceil$ is the ceiling function. Big $O$ and other related notations are used in the usual way.*

**Theorem 3** (Coverage upper bound, informal). *If we further assume that $Y$ is continuous, resulting in distinct prediction errors, then we also have the following upper bound:*

$$\mathbb{P}\left(|Y - \mathsf{RF}_B(X)| \leqslant \text{the } \lceil (n+1)(1-\alpha) \rceil \text{-th smallest } R_i - \varepsilon_{n,B}\right) \leqslant 1 - \alpha + \frac{1}{n+1} + O(\sqrt{\nu_{n,B}}).$$

As we detail below, the PIs we provide coverage guarantees for are neither the jackknife-with-stability interval discussed in [3], nor the jackknife+-after-bootstrap interval established in [18]. In our context, constructing the former needs $n$ leave-one-out (LOO) predictors (rather than $n$ OOB predictors), i.e., $n$ additional RFs with each built on a training set of size $n-1$. Constructing the latter needs the explicit information of each $\mathsf{RF}_B^{\backslash i}(\cdot)$ rather than the OOB prediction $\mathsf{RF}_B^{\backslash i}(X_i)$ for each $X_i$ only. Both these methods require additional, sometimes extensive, computation given current popular packages. In contrast, our results are operationally more convenient. After one-shot training, we obtain not only a point predictor $\mathsf{RF}_B(\cdot)$, but also a valid interval predictor at almost no extra cost. Under reasonable conditions, our results indicate that by slightly inflating (or deflating) the PI constructed from the $\lceil (n+1)(1-\alpha) \rceil$-th smallest $R_i$, the coverage probability is guaranteed not to decrease (or increase) too much from the desired level of $1 - \alpha$. In fact, many numerical results, such as those in [17, 35], suggest that

$$\mathbb{P}\left(|Y - \mathsf{RF}_B(X)| \leqslant \text{the } \lceil (n+1)(1-\alpha) \rceil \text{-th smallest } R_i\right) \approx 1 - \alpha.$$

Motivated by this fact, we further establish an asymptotic result of coverage for such PIs.

---

[1] When $\alpha \in (0, 1/(n+1))$, we follow the convention that the $(n+1)$-th smallest $R_i$ is $\infty$.

**Theorem 4.** *(Asymptotic coverage, informal) In addition to the conditions in the above theorems, also suppose the prediction error $|Y - \mathsf{RF}_B(X)|$ is continuous, and its cumulative distribution function (CDF) does not change too drastically for all sufficiently large $n$. Then*

$$\mathbb{P}\left(|Y - \mathsf{RF}_B(X)| \leqslant \text{ the } \lceil(n+1)(1-\alpha)\rceil\text{-th smallest } R_i\right) \to 1 - \alpha \text{ as } n \to \infty.$$

In [35], this asymptotic coverage was proved based on stronger assumptions. In particular, the true model is assumed to be additive such that "$Y = f_0(X) + $ noise" with the zero-mean noise independent of $X$, and $\mathsf{RF}_B(X)$ is assumed to converge to $f_0(X)$ in probability. We do not require $\mathsf{RF}_B$ to converge to anything in any sense when $n \to \infty$. Technically, we need the family of prediction error CDFs be uniformly equicontinuous.

Based on our results, the RF seems to be the only one, among existing popular machine learning algorithms, that can provide both point and interval predictors with justification in such a convenient way. This makes the RF appealing, especially for tasks where the computational cost is a concern.

It is also worth noting that the upper-bound result is of interest in its own right. It can be generalized to jackknife PIs that are constructed from any stable algorithm; the result serves as a complement to the lower bounds established previously [3, 18].

Summarizing, we

- theoretically prove that the (greedy) RF algorithm is stable when $Y^2$ does not have a heavy tail;
- numerically show that RF stability may hold beyond the above light-tail assumption;
- construct PIs based on the OOB error with finite-sample coverage guarantees: the lower bound of coverage does not need any additional assumption beyond stability; the upper bound needs an additional assumption, which is usually satisfied when $Y$ is continuous;
- provide the upper bound of coverage for jackknife PIs constructed from general stable algorithms, assuming distinct LOO errors; and
- prove asymptotically exact coverage for RF-based PIs under weaker assumptions than those previously considered in published work.

## 2 Concepts of algorithmic stability

Stability stands at the core of this work. There are different types of stability, each of which is used to assess quantitatively how stable (in some certain sense) an algorithm is with respect to small variations in training data [9, 28, 4]. In a recent work [4], robust optimization is used to enhance the stability of algorithms in classification tasks. In [28], bagging is proved to be an efficient mechanism to stabilize algorithms in regression tasks. We focus on regression here. As will be made clear, the technique used in this work relies on the fact that the RF predictor in regression results from averaging tree predictors. However, the majority vote of tree predictors is used in classification, and new ideas are needed to analyze the RF stability in this setting. For our purposes, we introduce three levels of stability from strongest to weakest. The strongest version of stability, introduced in [28], does *not* depend on the data distribution, and may be referred to as "absolute stability."

**Definition 1** (Absolute stability of algorithms). For any dataset consisting of $n \geqslant 2$ training points $D = \{(X_1, Y_1), \ldots, (X_n, Y_n)\}$ and any test point $(X, Y)$, an algorithm $\mathcal{A}$ is defined to be $(\varepsilon, \delta)$-absolutely-stable if

$$\frac{1}{n} \sum_{i=1}^{n} \mathbb{P}_\xi\left(\left|\hat{f}(X) - \hat{f}^{-i}(X)\right| > \varepsilon\right) \leqslant \delta$$

for some $\varepsilon, \delta \geqslant 0$, where $\xi$ denotes the possible innate randomness in the algorithm (such as the node splitting procedure in the RF) and can be seen as a random variable uniformly distributed in $[0, 1]$, $\hat{f} = \mathcal{A}(D; \xi)$ is the predictor trained on $D$, and $\hat{f}^{-i} = \mathcal{A}(D^{-i}; \xi)$ is the $i$th LOO predictor trained on $D^{-i}$, i.e., $D$ without the $i$th point $(X_i, Y_i)$. We might refer to the RF as both an algorithm (the learning procedure) and a predictor (the learned function) for simplicity.

Many bagged algorithms, in particular those with bounded predicted values, can achieve absolute stability with both $\varepsilon$ and $\delta$ converging to 0, as long as $n$ and the number of bags $B$ go to infinity.

However, the predicted value of the RF is in general unbounded (for regression tasks considered in this work), and we are more interested in another type of stability, investigated in [9], and called out-of-sample stability [3]. For simplicity, we name it "stability." This notion of stability turns out to be important in validating a jackknife prediction interval.

**Definition 2** (Stability of algorithms). For iid training and test data, algorithm $\mathcal{A}$ is $(\varepsilon, \delta)$-stable if

$$\mathbb{P}_{D,X,\xi}\left(\left|\hat{f}(X) - \hat{f}^{-i}(X)\right| > \varepsilon\right) \leqslant \delta$$

for some $\varepsilon, \delta \geqslant 0$, where $D, X, \hat{f}, \hat{f}^{-i}$ are as defined above.

We will establish this type of stability for the derandomized RF defined below, where the data-generating distribution is involved. To this end, we will use the methods in [28], which aim to provide absolute stability for bagged algorithms. Technically, we use such methods to first establish the "conditional stability" of an algorithm with respect to given data.

**Definition 3** (Conditional stability of algorithms). Conditional on $D$ and $X$, an algorithm $\mathcal{A}$ is defined to be $(\varepsilon, \delta)$-conditionally-stable if

$$\frac{1}{n}\sum_{i=1}^{n} \mathbb{P}_{\xi|D,X}\left(\left|\hat{f}(X) - \hat{f}^{-i}(X)\right| > \varepsilon\Big|D, X\right) \leqslant \delta$$

for some $\varepsilon, \delta \geqslant 0$, where $D, X, \hat{f}, \hat{f}^{-i}$ are as defined above.

Once conditional stability is established for the derandomized RF algorithm, its stability can be consequently established by invoking

$$\mathbb{P}_{D,X,\xi}(\cdot) = \mathbb{E}_{D,X}\left[\mathbb{P}_{\xi|D,X}(\cdot|D, X)\right].$$

Stability of the derandomized RF provides the most essential ingredient for that of the practical RF, although the latter involves another type of stability, known as ensemble stability [18]. Ensemble stability justifies replacing the LOO predictor with the OOB predictor in (1). We may abuse the term "stability" in the following when the OOB, rather than the LOO, predictor is used.

## 3 Stability of random forests

### 3.1 Basics of random forests

This work mainly considers using the RF to perform regression tasks, where the response $Y \in \mathbb{R}$ can be unbounded. By construction, the RF predictor with $B$ bags, denoted by $\mathsf{RF}_B$, is a bagged algorithm with the base algorithm being a tree, and $\mathsf{RF}_B = \frac{1}{B}\sum_{b=1}^{B}\mathsf{TREE}_b$, where $\mathsf{TREE}_b$ is the $b$th tree predictor, trained on the $b$th bag $r_b$, a bootstrapped sample of the training set $D$. The randomness in the tree predictor $\mathsf{TREE}$ originates from two independent sources: innate randomness $\xi$ in the node splitting process and resampling randomness from the bag $r$. For the $i$th point, one can define the OOB RF predictor as $\mathsf{RF}_B^{\setminus i} = \frac{1}{B_i}\sum_{b=1}^{B}\mathsf{TREE}_b \times \mathbb{I}\{i \notin r_b\}$, where $\mathbb{I}\{\cdot\}$ denotes the indicator function, and $B_i = \sum_{b=1}^{B}\mathbb{I}\{i \notin r_b\}$. Define $p \equiv \mathbb{P}(i \in r)$ as the probability that the $i$th point is included in bag $r$. Then it is clear that $B_i \sim \mathrm{Binomial}(B, 1-p)$ for all $i$. We also denote $\mathsf{rf}$ and $\mathsf{rf}^{\setminus i}$ as the *derandomized* versions of $\mathsf{RF}_B$ and $\mathsf{RF}_B^{\setminus i}$, respectively. Precisely, $\mathsf{rf} = \mathbb{E}_{\xi,r}[\mathsf{TREE}]$ and $\mathsf{rf}^{\setminus i} = \mathbb{E}_{\xi,r}[\mathsf{TREE}|i \notin r]$. It is worth noting that, by definition, $\mathsf{RF}_B^{\setminus i} \neq \mathsf{RF}_B^{-i}$ for finite $B$, while $\mathsf{rf}^{\setminus i} = \mathsf{rf}^{-i}$ as the derandomized RF results from the aggregation of an infinite number of trees. Since RF predictors are averages over tree predictors, the predicted values they output, given training set $D$, are bounded in $[Y_{(1)}, Y_{(n)}]$, where $Y_{(1)}$ and $Y_{(n)}$ are the minimum and maximum of $\{Y_1, \ldots, Y_n\}$, respectively. We also let $Z_i = |Y_i|$ for all $i$, and denote the maximum as $Z_{(n)}$. As a result, we have that

$$|\mathsf{rf}^{\setminus i} - \mathsf{rf}| \leqslant Y_{(n)} - Y_{(1)} \leqslant 2Z_{(n)}. \tag{2}$$

**Remark 1.** This is also true for $\mathsf{RF}_B^{\setminus i}$ and $\mathsf{RF}_B$ for any finite $B$. In fact, this is a distinctive feature of the RF, *irrespective* of the node splitting rule. Other regression methods do not necessarily have such a data-dependence bound. This observation helps to establish the conditional stability of the RF.

**Remark 2.** Practically, when $n$ is large, one might think that the bound (2) is crude. On one hand, if we look for a bound valid for any finite $n \geqslant 2$, then there is not much room for improvement for small $n$. On the other hand, we do expect that the *typical* stability of the RF can go beyond the finite-sample guarantee provided by (2) when $n$ is big, which is consistent with the numerical results shown below. A more informative bound for large $n$ is worth future investigation.

There are several quantities that are useful in establishing the RF stability; they can be calculated explicitly and are listed below. First, it is well known that

$$p \equiv \mathbb{P}(i \in r) = 1 - (1 - 1/n)^n = 1 - 1/e + O(1/n). \tag{3}$$

Actually, $p$ is monotonically decreasing for $n \geqslant 1$. Second,

$$q \equiv -\mathrm{Cov}(\mathbb{I}\{i \in r\}, \mathbb{I}\{j \in r\}) = (1 - 1/n)^{2n} - (1 - 2/n)^n = O(1/n), \tag{4}$$

as can be directly checked. Third, the moment generating function of $B_i$ is

$$\mathbb{E}\left[e^{sB_i}\right] = (p + (1 - p)e^s)^B. \tag{5}$$

In the following, we first perform the stability analysis for the derandomized RF (consisting of an infinite number of trees) and then extend the results to the practical finite-$B$ case.

### 3.2 Derandomized random forests

The following theorem formalizes the conditional stability property for the derandomized RF, the proof of which is a direct result of Theorem 8 in [28], and is omitted here.

**Theorem 5** (Conditional stability of derandomized random forests)**.** *Conditional on training set $D$ and test point $(X, Y)$, for the derandomized random forest predictor* rf *we have that*

$$\frac{1}{n}\sum_{i=1}^{n}\mathbb{I}\left\{\left|\mathrm{rf}(X) - \mathrm{rf}^{\backslash i}(X)\right| > \varepsilon \Big| D, X\right\} \leqslant \delta(D, X) \equiv \frac{Z_{(n)}^2}{\varepsilon^2 n}\left(\frac{p}{1 - p} + \frac{q}{(1 - p)^2}\right). \tag{6}$$

If $\delta(D, X) \geqslant 1$, the statement is trivial, and we will focus on the case that $\delta(D, X) \in (0, 1)$. We can now establish the stability property for the derandomized RF.

**Theorem 6** (Stability of derandomized random forests)**.** *For iid training and test data and $\varepsilon > 0$, the derandomized random forest predictor* rf *is stable with*

$$\mathbb{P}_{D,X}\left(\left|\mathrm{rf}(X) - \mathrm{rf}^{\backslash i}(X)\right| > \varepsilon\right) \leqslant \frac{\mathbb{E}[Z_{(n)}^2]}{\varepsilon^2 n}\left(\frac{p}{1 - p} + \frac{q}{(1 - p)^2}\right) \equiv \nu. \tag{7}$$

This result follows directly from the conditional stability (6) by averaging over $D$ and $X$. There is some freedom in choosing the dependence of $\varepsilon$ on $n$. On one hand, in order to make sense of the word "stability," we do expect $\varepsilon$ and $\nu$ to be small for large $n$. From (3) and (4), it is clear that the asymptotic behavior of $\nu$ is dominated by that of $\mathbb{E}[Z_{(n)}^2]/(\varepsilon^2 n)$, which can be tuned by manipulating $\varepsilon$. For example, a matching convergence rate to 0 between $\varepsilon$ and $\nu$ might be desirable, and one can then set $\varepsilon = O((\mathbb{E}[Z_{(n)}^2]/n)^{1/3})$ if the scaling of $\mathbb{E}[Z_{(n)}^2] = o(n)$ is known or can be inferred. On the other hand, we can fix $\varepsilon$ to further investigate the relation between stability and the convergence-in-probability property of the RF. By (7), under the condition that $\mathbb{E}[Z_{(n)}^2]/n \to 0$ as $n \to \infty$, one immediately comes to the conclusion that $\mathrm{rf}^{\backslash i}(X) - \mathrm{rf}(X)$ converges to 0 in probability. Actually, a stronger conclusion can be drawn under the same condition.

**Corollary 1.** *For iid training and test data, we have*

$$\mathbb{E}_{D,X}\left[|\mathrm{rf}(X) - \mathrm{rf}^{\backslash i}(X)|\right] < 2\sqrt{\frac{\mathbb{E}[Z_{(n)}^2]}{n}\left(\frac{p}{1 - p} + \frac{q}{(1 - p)^2}\right)}. \tag{8}$$

*Further assume that $\mathbb{E}[Y^2] < \infty$. Then we have*

$$\mathbb{E}_{D,X}\left[|\mathrm{rf}(X) - \mathrm{rf}^{\backslash i}(X)|\right] \to 0 \text{ and } \mathrm{rf}(X) - \mathrm{rf}^{\backslash i}(X) \xrightarrow{\mathbb{P}} 0 \text{ as } n \to \infty. \tag{9}$$

**Remark 3.** The additional assumption that $\mathbb{E}[Y^2] < \infty$ is mild. Many commonly encountered random variables have a light tail and thus a finite second moment, irrespective of the detailed information of the distribution in question. Note that the bound (2) itself can be crude, and our result is expected to be valid even beyond this mild condition.

**Remark 4.** This result indicates that the *difference* is diminishing between rf and $\mathrm{rf}^{\backslash i}$, built on $n$ and $n-1$ training data points, respectively. However, there is no indication that the derandomized $\mathrm{rf}(X)$ itself will converge to anything. This idea inspires the proposal of Theorem 11.

The proof of this result, as well as others below, will be deferred to the Appendix. So far, we have investigated the derandomized version of the RF, which is a limiting case and can be seen as consisting of an infinite number of trees, averaging out all kinds of possible randomness in the predictor construction process. In order to make the results more relevant to applied machine learning, the finite-$B$ analysis for the RF is conducted below.

### 3.3 Finite-$B$ random forests

We now consider the difference between $\mathsf{RF}_B$ and $\mathsf{RF}_B^{\backslash i}$. We denote $\boldsymbol{\xi} = (\xi_1, \ldots, \xi_B)$ and $\boldsymbol{r} = (r_1, \ldots, r_B)$ as the corresponding sources of randomness in $B$ trees. We also consider conditional stability first and then move to the stability of $\mathsf{RF}_B$.

**Theorem 7** (Conditional stability of finite-$B$ random forests). *Conditional on training set $D$ and test point $(X, Y)$, for a random forest predictor $\mathsf{RF}_B$ that consists of $B$ trees, we have for $\varepsilon > 0$ that*

$$\frac{1}{n}\sum_{i=1}^{n} \mathbb{P}_{\boldsymbol{\xi},\boldsymbol{r}|D,X}\left(\left|\mathsf{RF}_B(X) - \mathsf{RF}_B^{\backslash i}(X)\right| > \varepsilon + 2\sqrt{\frac{2Z_{(n)}^2}{B}\ln\left(\frac{1}{\delta}\right)}\middle| D, X\right) \leqslant 3\delta + g(p, \delta, B),$$

*where $\delta$ is short for $\delta(D, X)$ as defined in (6) and $g(p, \delta, B) = 2(p + (1-p)\delta^{\frac{1}{B}})^B$.*

Next, we consider the case of iid data and investigate the RF stability by averaging out the randomness in data. Note that $Z_{(n)}$ and $\delta$ are random and depend on the data distribution, while we are interested in a probability bound for $|\mathsf{RF}_B(X) - \mathsf{RF}_B^{\backslash i}(X)|$ greater than a deterministic quantity, which is only a function of $B$ and $n$. In this finite-$B$ case, the stability of $\mathsf{RF}_B$ cannot be directly obtained from its conditional stability as in the derandomized situation.

**Theorem 8** (Stability of finite-$B$ random forests). *Assume training points in set $D$ and the test point $(X, Y)$ are iid, drawn from a fixed distribution. For the random forest predictor $\mathsf{RF}_B$ consisting of $B$ trees and trained on $D$, we have*

$$\mathbb{P}_{D,X,\boldsymbol{\xi},\boldsymbol{r}}\left(\left|\mathsf{RF}_B(X) - \mathsf{RF}_B^{\backslash i}(X)\right| > \varepsilon_{n,B}\right) \leqslant \nu_{n,B}, \tag{10}$$

*where $\varepsilon_{n,B} = \sum_{i=1}^{3}\varepsilon_i$, and $\nu_{n,B} = \sum_{i=1}^{3}\nu_i$. The pair of $(\varepsilon_2, \nu_2/\lambda)$ satisfies the derandomized RF stability condition (7) with $\lambda > 1$. Moreover, $\varepsilon_1 = \varepsilon_3 = \sqrt{2\lambda\mathbb{E}[Z_{(n)}^2]\ln(\frac{1}{\nu_2})/B}$, $\nu_1 = 2\nu_2 + 2\mathbb{P}(Z_{(n)}^2 > \lambda\mathbb{E}[Z_{(n)}^2])$, and $\nu_3 = g(p, \nu_2, B) + 2\mathbb{P}(Z_{(n)}^2 > \lambda\mathbb{E}[Z_{(n)}^2])$.*

On a high level, the establishment of this theorem relies on two observations: (i) the stability of the derandomized RF, so that the difference $|\mathrm{rf}(X) - \mathrm{rf}^{\backslash i}(X)|$ is controlled, and (ii) the concentration of measure, so that the differences $|\mathsf{RF}_B(X) - \mathrm{rf}(X)|$ and $|\mathsf{RF}_B^{\backslash i}(X) - \mathrm{rf}^{\backslash i}(X)|$ are controlled. In order to make full sense of the word "stability," it is desirable that $\varepsilon_{n,B}$ and $\nu_{n,B}$ can converge to 0. It is known that $\mathbb{E}[Y^2] < \infty$ suffices to ensure $\mathbb{E}[Z_{(n)}^2] = o(n)$ [15, 13], and hence the stability of the derandomized RF. Now in the finite-$B$ case, we need an additional distributional assumption to control the tail probability $\mathbb{P}(Z_{(n)}^2 > \lambda\mathbb{E}[Z_{(n)}^2])$. It turns out that for typical light-tailed $Y^2$, such a tail probability will converge to 0 as $n \to \infty$. Technically, we can assume $Y^2$ to be sub-gamma [8]. Note that bounded and sub-Gaussian random variables are sub-gamma. Hence the sub-gamma assumption is not strong and can be satisfied by distributions underlying many real datasets.

**Definition 4** (Sub-gamma random variables [8]). A random variable $W$ is said to be sub-gamma (on the right tail) with parameters $(\sigma^2, c)$ where $c \geqslant 0$, if $\ln\mathbb{E}[e^{s(W-\mathbb{E}[W])}] \leqslant \frac{s^2\sigma^2}{2(1-cs)}$ for all $s \in (0, 1/c)$.

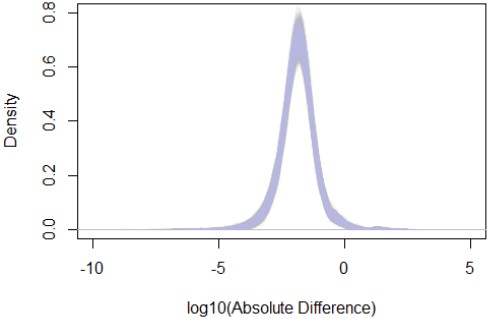 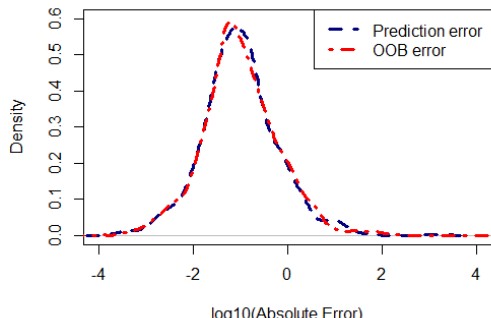

Figure 1: Left: Density plots of the $\log_{10}$ absolute difference $|\mathsf{RF}_B(X) - \mathsf{RF}_B^{\backslash i}(X)|$ for 3000 OOB predictors $\mathsf{RF}_B^{\backslash i}$ on 1000 test points. We let $B = 1000$. The RF stability (10) seems to persist, even though $Y$ follows the (heavy-tailed) standard Cauchy distribution. Numerically, we set $\hat{\nu}_{n,B} = 0.05$ and calculated the maximum of the 0.95 quantile of the 3000 empirical distributions to have $\hat{\varepsilon}_{n,B} = 0.237$. Right: Density plots of 1000 $\log_{10}$ absolute prediction errors $|Y - \mathsf{RF}_B(X)|$ and of 3000 $\log_{10}$ absolute OOB errors $|Y_i - \mathsf{RF}_B^{\backslash i}(X_i)|$. The similarity between the plots supports the idea that the OOB errors can be used to construct PIs.

**Lemma 1.** *Suppose $Y^2$ is sub-gamma with parameters $(\sigma^2, c)$ with $c > 0$, and $\mathbb{E}[Z_{(n)}^2] \sim a \ln n$ with $a \leqslant c$. For $\lambda > c/a$, we have $\lim_{n \to \infty} \mathbb{P}(Z_{(n)}^2 > \lambda \mathbb{E}[Z_{(n)}^2]) = 0$.*

**Remark 5.** We have set $c > 0$ above. If $c = 0$, then $Y^2$ is in fact sub-Gaussian, and the tail probability can be controlled similarly. If $Y^2$ is upper bounded by some constant $M^2$, the stability analysis is even simpler, and there is no need to consider the tail probability at all, as we can use $M^2$ in place of $Z_{(n)}^2$ in the conditional stability of the RF and then take expectation with respect to data.

**Example 1.** *Consider $Y^2 \sim \mathsf{Exp}(1)$, the exponential distribution with scale parameter 1. It is known that $Y^2$ is sub-gamma with $(\sigma^2, c) = (1, 1)$ [8], and $\mathbb{E}[Z_{(n)}^2] = \sum_{i=1}^n \frac{1}{i} \equiv H_n$ with $H_n \in (\gamma + \ln n, \gamma + \ln(n + 1))$, where $\gamma \approx 0.577$ is Euler's constant. Hence $H_n = \ln n + o(\ln n)$, and a straightforward calculation reveals that $\lim_{n \to \infty} \mathbb{P}(Z_{(n)}^2 > \lambda \mathbb{E}[Z_{(n)}^2]) = 0$ as long as $\lambda > 1$.*

From such results, one can see that the vanishing tail probability is not a stringent condition. By taking this additional assumption, it is indeed possible that both $\varepsilon_{n,B}$ and $\nu_{n,B}$ converge to 0.

**Corollary 2.** *For the same setting as in Theorem 8, suppose $Y^2$ is sub-gamma with parameters $(\sigma^2, c)$ with $c > 0$ and $\mathbb{E}[Z_{(n)}^2] \sim a \ln n$ with $a \leqslant c$. Let $\lambda > c/a$ be a fixed number, and let $B$ depend on $n$. Then for $\varepsilon_2$ that satisfies both $\varepsilon_2 = \omega(\sqrt{\ln n/n})$ and $\varepsilon_2 = o(1)$, and $B = \Omega(\ln^2 n)$, we have $\lim_{n \to \infty} \varepsilon_{n,B} = \lim_{n \to \infty} \nu_{n,B} = 0$.*

It is worth noting that there are multiple ways to let $\varepsilon_{n,B}$ and $\nu_{n,B}$ approach 0, as the dependence of $\varepsilon_2$, $B$, and even $\lambda$ on $n$ can all be manipulated. The point is that, theoretically, even the greedy RF can be stable with vanishing parameters. In practice, however, the stability of $\mathsf{RF}_B$ seems to hold in broader situations where both the moment and tail assumptions on $Y^2$ can be relaxed.

### 3.4 Stability in practice and limitations of theory

We created a virtual dataset consisting of $n = 4000$ points. We let $Y$ be a standard Cauchy random variable, which is even without a well-defined mean. The feature vector $X \in \mathbb{R}^3$ is determined as $X = [0.5Y + \sin(Y), Y^2 - 0.2Y^3, \mathbb{I}\{Y > 0\} + \zeta]^T$ where $\zeta$ is a standard normal random variable. We used 3000 of the points for training and 1000 of them as test points. Using the `randomForest` package with default setting (except letting $B = 1000$), we had an output RF predictor $\mathsf{RF}_B$. We also aggregated corresponding tree predictors to have 3000 OOB predictors $\mathsf{RF}_B^{\backslash i}$. For each $i \in [3000]$, we calculated the absolute difference $|\mathsf{RF}_B(X) - \mathsf{RF}_B^{\backslash i}(X)|$ on 1000 test points to come up with a

density plot for such a difference, shown in Fig. 1. We also calculated 1000 absolute prediction errors $|Y - \mathsf{RF}_B(X)|$ that are incurred by $\mathsf{RF}_B$ on test points, and 3000 OOB errors $|Y_i - \mathsf{RF}_B^{\backslash i}(X_i)|$, each incurred by an OOB predictor $\mathsf{RF}_B^{\backslash i}$ on its OOB point $(X_i, Y_i)$. The computation can be done within a few minutes on a laptop. The density plots of these two kinds of errors are also shown in Fig. 1. This example shows that the RF stability can be present beyond the realm guaranteed by the light-tail assumption. As mentioned above, this is because the bound (2) can be conservative when $n$ is large. We hope our results can inspire future study towards a more informative bound. Also, the similarity between the prediction error and the OOB error in this heavy-tail case indicates that the RF-based PIs analyzed below can find more applications in practice than justified by the current theory.

## 4  Random-forest prediction intervals

### 4.1  Comparison with related methods

With the stability property of the RF, it is possible to construct PIs with finite-sample guaranteed coverage. Recent years have witnessed the development of distribution-free predictive inference [1] with the full [33, 27], split [24, 31, 20], and jackknife+ [3, 32] conformal prediction methods being three milestones. The full conformal method is computationally prohibitive when used in practice. The split method greatly reduces the computational cost but fails to thoroughly extract the available information of training data. The jackknife+ (**J+**) method maximizes the usage of data at a computational cost in between those of full and split methods. In [18], jackknife+-after-bootstrap (**J+aB**) was proposed for bagged algorithms to achieve the same goal as in J+, while the training cost can be further reduced. However, the number of bags $B$ is required to be a Binomial random variable, which might seem unnatural. It turns out that by further imposing the assumption of *ensemble* stability (which is essentially the concentration of resampling measure), J+aB can still have guaranteed coverage with a fixed $B$. Ensemble stability is defined for bagged algorithms. It measures how typical a bootstrap sample is, and is different from the algorithmic stability that quantifies the influence of removing one training point. If algorithmic stability is also imposed, then not only J+aB, but also jackknife can provide guaranteed coverage, which is otherwise impossible [29, 3].

Conceptually, the J+ approach and its variants under stability conditions are particularly relevant to this work. As the stability we establish for the RF contains both ensemble and algorithmic components, we will generally refer to the J+aB method with both ensemble and algorithmic stability as **J+aBS** and the jackknife method with algorithmic stability as **JS**. Our method might be best described as "jackknife-after-bootstrap-with-stability (**JaBS**)" tailored for the RF, which is different from both JS and J+aBS. Our method requires the least effort of computing as only one output predictor is needed, while all others require at least $n$ output predictors.

There also exist RF-based PIs [17, 35] that are essentially of the jackknife-after-bootstrap (**JaB**) type and almost identical to ours practically when $\varepsilon$ is small and $n$ equals the size of a typical dataset. However, without stability, there is, in general, no guarantee for the coverage of such PIs, although the asymptotic coverage $1 - \alpha$ can be established based on strong assumptions [35]. We take advantage of the stability of the RF algorithm to establish the lower bound of coverage in Theorem 9 below. An upper bound is established in Theorem 10 with an additional mild assumption. We also propose a weaker assumption for asymptotic coverage in Theorem 11.

We compare these relevant methods to ours in Table 1 and Table 2, where the RF is set as the working algorithm for all methods and $(\varepsilon, \nu)$ is a general pair of stability parameters. We define $q_{n,\alpha}\{R_i\}$, $q_{n,\alpha}^+\{R_i\}$, $q_{n,\alpha}^-\{R_i\}$, and $q_{n,\alpha}'\{R_i\}$ as follows. Given $\{a_1, \ldots, a_n\}$,

$$q_{n,\alpha}\{a_i\} = q_{n,\alpha}^+\{a_i\} \equiv \text{the } \lceil(1-\alpha)(n+1)\rceil\text{-th smallest value of } \{a_1, \ldots, a_n\},$$

$$q_{n,\alpha}'\{a_i\} \equiv \text{the } \lceil(1-\alpha)n\rceil\text{-th smallest value of } \{a_1, \ldots, a_n\},$$

$$q_{n,\alpha}^-\{a_i\} \equiv \text{the } \lfloor\alpha(n+1)\rfloor\text{-th smallest value of } \{a_1, \ldots, a_n\},$$

where $\lfloor \cdot \rfloor$ is the floor function. Let $R_i^{\mathrm{LOO}} = |Y_i - \mathsf{RF}_B^{-i}(X_i)|$ be the LOO error, where $\mathsf{RF}_B^{-i}$ is trained without the $i$th training point, and by definition $\mathsf{RF}_B^{-i} \neq \mathsf{RF}_B^{\backslash i}$.

In Table 1, we list the corresponding PI constructed from each method and the output predictors of each method. The number of output predictors directly reflects the computational cost. It is

Table 1: Methods to construct prediction intervals using random forests: computational cost

| Method | Output predictors | Prediction interval for future $Y$ |
|---|---|---|
| J+ [3] | $\mathsf{RF}_B^{-i}, i \in [n]$ | $[q_{n,\alpha}^-\{\mathsf{RF}_B^{-i}(X) - R_i^{\mathrm{LOO}}\}, q_{n,\alpha}^+\{\mathsf{RF}_B^{-i}(X) + R_i^{\mathrm{LOO}}\}]$ |
| J+aB [18] | $\mathsf{RF}_B^{\backslash i}, i \in [n]$ | $[q_{n,\alpha}^-\{\mathsf{RF}_B^{\backslash i}(X) - R_i\}, q_{n,\alpha}^+\{\mathsf{RF}_B^{\backslash i}(X) + R_i\}]$ |
| JS [3] | $\mathsf{RF}_B$ and $\mathsf{RF}_B^{-i}, i \in [n]$ | $\mathsf{RF}_B(X) \pm q_{n,\alpha}\{R_i^{\mathrm{LOO}} + \varepsilon\}$ |
| J+aBS [18] | $\mathsf{RF}_B^{\backslash i}, i \in [n]$ | $[q_{n,\alpha}^-\{\mathsf{RF}_B^{\backslash i}(X) - R_i\} - \varepsilon, q_{n,\alpha}^+\{\mathsf{RF}_B^{\backslash i}(X) + R_i\} + \varepsilon]$ |
| JaB | $\mathsf{RF}_B$ | $\mathsf{RF}_B(X) \pm q_{n,\alpha}\{R_i\}$ [17] |
| | | $\mathsf{RF}_B(X) \pm q'_{n,\alpha}\{R_i\}$ [35] |
| Ours (JaBS) | $\mathsf{RF}_B$ | $\mathsf{RF}_B(X) \pm q_{n,\alpha}\{R_i + \varepsilon\}$ (Theorem 9) |
| | | $\mathsf{RF}_B(X) \pm q_{n,\alpha}\{R_i - \varepsilon\}$ (Theorem 10) |
| | | $\mathsf{RF}_B(X) \pm q_{n,\alpha}\{R_i\}$ (Theorem 11) |

Table 2: Methods to construct prediction intervals using random forests: theoretical coverage

| Method | Theoretical coverage | Additional conditions |
|---|---|---|
| J+ [3] | $\geqslant 1 - 2\alpha$ | None |
| J+aB [18] | $\geqslant 1 - 2\alpha$ | Binomial $B$ |
| JS [3] | $\geqslant 1 - \alpha - O(\sqrt{\nu})$ | Stability (algorithmic) |
| J+aBS [18] | $\geqslant 1 - \alpha - O(\sqrt{\nu})$ | Stability (ensemble + algorithmic) |
| JaB | No guarantee [17] | - |
| | $\to 1 - \alpha$ [35] | Strong (additive model, consistency of RF predictor) |
| Ours (JaBS) | $\geqslant 1 - \alpha - O(\sqrt{\nu})$ | Stability (Theorem 9) |
| | $\leqslant 1 - \alpha + \frac{1}{n+1} + O(\sqrt{\nu})$ | + Distinct residuals (Theorem 10) |
| | $\to 1 - \alpha$ | + Uniformly equicontinuous CDF of $|Y - \mathsf{RF}_B(X)|$ and vanishing $\varepsilon, \nu$ (Theorem 11) |

worth noting that acquiring the LOO predictor $\mathsf{RF}_B^{-i}$ needs substantial computation. In packages like `randomForest`, aggregating tree predictors to obtain the OOB predictor $\mathsf{RF}_B^{\backslash i}$ also needs extra computation. However, the predicted value $\mathsf{RF}_B^{\backslash i}(X_i)$ can be obtained immediately by calling the predict() function. The fact that the value of $\mathsf{RF}_B^{\backslash i}(X)$ on a test point is *not* needed further reduces the computational cost of JaB and our method, which only need one output RF predictor, and are more favorable computationally.

In Table 2, we list the coverage of the PI constructed from each method, as well as the additional conditions (beyond iid data) needed to achieve the coverage. Note that J+ does not require any additional conditions to achieve the coverage lower bound $1 - 2\alpha$, but J+aB requires that the number of trees $B$ be a Binomial random variable. For JS, J+aBS, and our method, stability is needed to achieve the coverage lower bound $1 - \alpha - O(\sqrt{\nu})$. With additional mild assumptions, the coverage upper bound and asymptotic coverage of our method can be established. However, there is no guarantee of coverage for JaB without strong assumptions.

In summary, our theoretical work provides a series of coverage guarantees to a computationally feasible method for constructing PIs based on the RF algorithm. In the following, we will establish the lower and upper bound of coverage, as well as the asymptotic coverage.

## 4.2   Non-asymptotic coverage guarantees

**Theorem 9** (Coverage lower bound). *Suppose the RF predictor* $\mathsf{RF}_B$ *satisfies the stability condition as in Theorem 8. Then we have for a test point* $(X, Y)$ *that*

$$\mathbb{P}(Y \in \mathsf{RF}_B(X) \pm q_{n,\alpha}\{R_i + \varepsilon_{n,B}\}) \geqslant 1 - \alpha - \nu_1 - 2\sqrt{\nu_2} - 2\sqrt{\nu_3}. \tag{11}$$

This result is established by starting from the analysis of an imaginary extended dataset $\overline{D} = D \cup \{(X, Y)\}$, where the test point is *assumed* to be known. We denote $(X, Y)$ as $(X_{n+1}, Y_{n+1})$ for convenience. For all points in $\overline{D}$, consider the derandomized RF predictor $\widetilde{\mathsf{rf}}^{\backslash i}$ that is built on $n$ data

points without the $i$th point in $\overline{D}$, $i \in [n+1]$. One can then define the OOB error $\widetilde{r}_i \equiv |Y_i - \widetilde{\mathsf{rf}}^{\backslash i}|$. Since all data are iid, we have that $\mathbb{P}(\widetilde{r}_{n+1} \leqslant q_{n,\alpha}\{\widetilde{r}_i\}) \geqslant 1 - \alpha$, where $q_{n,\alpha}\{\widetilde{r}_i\}$ is the $\lceil(1-\alpha)(n+1)\rceil$-th smallest value of $\{\widetilde{r}_1, \ldots, \widetilde{r}_n\}$. Next, notice $\widetilde{r}_{n+1} = |Y_{n+1} - \widetilde{\mathsf{rf}}^{\backslash(n+1)}(X_{n+1})| = |Y_{n+1} - \mathsf{rf}(X_{n+1})|$ by the definitions of $\widetilde{\mathsf{rf}}^{\backslash(n+1)}$ and $\mathsf{rf}$. By concentration of measure, $\mathsf{rf}(X_{n+1})$ can be approximated by $\mathsf{RF}_B(X_{n+1})$, and thus $\widetilde{r}_{n+1}$ can be roughly replaced with $|Y_{n+1} - \mathsf{RF}_B(X_{n+1})|$, which is desired. Then by stability of $\mathsf{rf}$, $\{\widetilde{r}_i\}$ can be approximated by $\{r_i \equiv |Y_i - \mathsf{rf}^{\backslash i}(X_i)|\}$. Although $\{r_i\}$ is still unavailable in practice, by applying the idea of concentration of measure again, $\{r_i\}$ can be further approximated by $\{R_i\}$, which is accessible given $D$. Eventually, we can bound $|Y_{n+1} - \mathsf{RF}_B(X_{n+1})|$ in terms of $\{R_i\}$. The approximations are accounted for by the stability parameters in Theorem 8.

If we further assume that there are no ties among $\{\widetilde{r}_i\}$, $i \in [n+1]$, a typical case when $Y$ is continuous, then we can also establish the upper bound of coverage.

**Theorem 10** (Coverage upper bound). *Suppose there are no ties in $\{\widetilde{r}_i\}$, $i \in [n+1]$, and the RF predictor $\mathsf{RF}_B$ satisfies the stability condition as in Theorem 8. Then*

$$\mathbb{P}(Y \in \mathsf{RF}_B(X) \pm q_{n,\alpha}\{R_i - \varepsilon_{n,B}\}) \leqslant 1 - \alpha + \frac{1}{n+1} + \nu_1 + 2\sqrt{\nu_2} + 2\sqrt{\nu_3}. \tag{12}$$

The upper bound can be established because if there are no ties among $\widetilde{r}_1, \ldots, \widetilde{r}_{n+1}$, then $\mathbb{P}(\widetilde{r}_{n+1} \leqslant q_{n,\alpha}\{\widetilde{r}_i\}) \leqslant 1 - \alpha + \frac{1}{n+1}$. The apparent symmetry between the lower and upper bound originates from the fact that they both are established by using the RF stability once and the concentration of measure twice. Note that this idea can be applied to JS intervals for an arbitrary stable algorithm in exactly the same way, providing a complement to the lower bound for JS intervals established in [3].

**Corollary 3** (Coverage upper bound for jackknife-with-stability intervals). *Let $\hat{f}$ be a predictor trained on $n$ iid data points and $\hat{f}^{-i}$ be the LOO predictor without the $i$th point. Suppose $\hat{f}$ is stable with $\mathbb{P}(|\hat{f}(X) - \hat{f}^{-i}(X)| > \varepsilon) \leqslant \nu$, and the LOO errors are distinct on the extended training set that includes an iid test point $(X,Y)$. Then we have $\mathbb{P}\left(|Y - \hat{f}(X)| \leqslant q_{n,\alpha}\{r_i - \varepsilon\}\right) \leqslant 1 - \alpha + \frac{1}{n+1} + 2\sqrt{\nu}$, where $r_i$ are the LOO errors on the original training set.*

### 4.3 Asymptotic coverage guarantee

As shown above, the stability parameters $(\varepsilon_{n,B}, \nu_{n,B})$ can vanish when $n \to \infty$. It is reasonable to expect that $\mathbb{P}(Y \in \mathsf{RF}_B(X) \pm q_{n,\alpha}\{R_i\}) \to 1 - \alpha$ in this limit, as is consistent with numerous empirical observations [17, 35]. However, to achieve this goal, it seems that more assumptions are unavoidable. In [35], the guaranteed coverage of the JaB method is established by assuming that $\mathsf{RF}_B(X)$ converges to some $f_0(X)$ in probability as $n \to \infty$, where $f_0$ is the true regression function of an additive model that generates the data. We show that this can be done under weaker conditions.

**Theorem 11** (Asymptotic coverage). *Denote $F_n$ as the CDF of $|Y - \mathsf{RF}_B(X)|$. Suppose $\{F_n\}_{n \geqslant n_0}$ is uniformly equicontinuous for some $n_0$. Then $\mathbb{P}(Y \in \mathsf{RF}_B(X) \pm q_{n,\alpha}\{R_i\}) \to 1 - \alpha$ as $n \to \infty$ when conditions in Theorem 9, Theorem 10, and Corollary 2 are satisfied.*

**Remark 6.** Intuitively, using errors from $\mathsf{RF}_B^{\backslash i}$ that are trained on $n - 1$ points to approximate those from $\mathsf{RF}_B$, trained on $n$ points, we only need this approximation to be exact asymptotically. There is no need for $\mathsf{RF}_B$ itself to converge to anything. This is one major conceptual difference between our work and [35], and it is in this sense that our assumption is weaker. Practically, this kind of PI is recommended as it does not involve $(\varepsilon_{n,B}, \nu_{n,B})$, and has great performance on numerous datasets.

## 5   Conclusion

In this work, for the first time, we theoretically establish the stability property of the greedy version of random forests, which is implemented in popular packages. The theoretical guarantee is based on a light-tail assumption of the marginal distribution of the squared response $Y^2$. However, numerical evidence suggests that this stability could persist in much broader situations. Based on the stability property and some mild conditions, we also establish finite-sample lower and upper bounds of coverage, as well as the exact coverage asymptotically, for prediction intervals constructed from the out-of-bag error of random forests, justifying random forests as an appealing method to provide both point and interval prediction simultaneously.

## Acknowledgments and Disclosure of Funding

Much of the work was completed while Yan Wang was a PhD student in the Department of Statistics at Iowa State University. This research was supported in part by the US National Science Foundation under grant HDR:TRIPODS 19-34884.

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
