# OpenReview forum: "Stability of Random Forests and Coverage of Random-Forest Prediction Intervals"
_NeurIPS.cc/2023/Conference — NeurIPS 2023 poster_

### Official Review · Reviewer_BF9X · 2023-07-06

**Soundness:** 2 fair
**Presentation:** 2 fair
**Contribution:** 2 fair
**Rating:** 2
**Confidence:** 5

**Summary:**

This work studies the random forests stablity for regression problem, and the authors presents theoretical analysis on the upper and lower boudns for the coverage probability of prediction intervals constructed from the out-of-bag error of random forests. The theoretical guarantee is based on a light-tail assumption of the marginal distribution of the squared response.

------------------after response---------------------------------

After reading the authors' response, I do not think the authors answer my concerns, in particularly for the novelties and significances.

As a theoreical work, it is very important to evaluate from theoretical novelties and techniques, while I find some inremental results based on well-known techniques.

I do not find any experiments, and I do not know why the authors could claim that "Our work applies to many variants of random forests ... which makes it particularly relevant in theory and practice".

**Strengths:**

1) It is an interesting problem on the theoretical understanding of random forests.

2) Some theoretical results on the convergence probability of prediction intervals constructed from the out-of-bag error of random forests.

3) Limited theoretical techinical contributions

**Weaknesses:**

1) The problem is not very clear. The authors should first present the studied problem, i.e. the original ranfom forests for regression, or randomf forest interval, or  prediction intervals constructed from the out-of-bag error of random forests. It is very confused to understand the main contributions in the current submission and relevant work. For completeness, it would be better to present the detailed algorithm, rather than finding some other research work for other readers.

2) The main conclusions are not clear. The main contribution of this work is the stability of random forests in Theorem 1. Generally, a theoretical work concerns seriously the convergence rate of stability, and how tight of this rate. It would be better to present the specific expression for \nu_{n,B}, and make necessary discussions. What's factors affect the stability rate?

3) Some important definitions and notions are missing. For example, where is the definition of "light tail"， whichi is the basic assumption in main theoretical results. How to characaterize light tail and its relevant factors.

4) The authros should clearify the novelty and significance of the main results, for example, how about the theoretical new insghts on the technical proof in this work. As an pure theoretical problem, it is importnat to present some new technical proof, rather than simple extension of the current techiniques. What's the siginificance of the main results, is it possible to present some pratical guidance and suggest some new algorithms.

5) The authors should have a good background on the theoretical analysis on random forests, for example,
G. Biau, L. Devroye, G. Lugosi, Consistency of random forests and other averaging classifiers, JMLR 2008.
M. Denil, D. Matheson, N. De Freitas, Narrowing the gap: random forests in theory and in practice, ICLM2014.
W. Gao, F. Xu and Z.-H. Zhou. Towards convergence rate analysis of random forests for classification. AIJ, 2022.

**Questions:**

1) The problem is not very clear. The authors should first present the studied problem, i.e. the original ranfom forests for regression, or randomf forest interval, or  prediction intervals constructed from the out-of-bag error of random forests. It is very confused to understand the main contributions in the current submission and relevant work. For completeness, it would be better to present the detailed algorithm, rather than finding some other research work for other readers.

2) The main conclusions are not clear. The main contribution of this work is the stability of random forests in Theorem 1. Generally, a theoretical work concerns seriously the convergence rate of stability, and how tight of this rate. It would be better to present the specific expression for \nu_{n,B}, and make necessary discussions. What's factors affect the stability rate?

3) Some important definitions and notions are missing. For example, where is the definition of "light tail"， whichi is the basic assumption in main theoretical results. How to characaterize light tail and its relevant factors.

4) The authros should clearify the novelty and significance of the main results, for example, how about the theoretical new insghts on the technical proof in this work. As an pure theoretical problem, it is importnat to present some new technical proof, rather than simple extension of the current techiniques. What's the siginificance of the main results, is it possible to present some pratical guidance and suggest some new algorithms.

**Limitations:**

This is a pure theoretical work

---

> ### Author Rebuttal · Authors · 2023-08-06
>
> We thank the reviewer for agreeing that "It is an interesting problem on the theoretical understanding of random forests." We will seriously take the reviewer's comments into consideration and make revisions accordingly. Below are point-by-point responses to the reviewer's "Weaknesses" comments.
> 1. We want to make it clear here that there are two major contributions of our work. The first is establishing the stability of random forests under mild conditions; the second is providing non-asymptotic coverage guarantees for random forest prediction intervals (RFPIs), which is based on the stability property. It is worth stressing that methods to construct RFPIs have been proposed in Refs. [14] and [32]. However, in Ref. [14], there is no theoretical guarantee for the coverage probability; in Ref. [32], the coverage probability is established under very strong assumptions and it is an asymptotic result. It is NOT our aim to provide a completely new way to construct RFPIs. Rather, we provide further theoretical justifications to existing methods under much weaker assumptions. To the best of our knowledge, our non-asymptotic guarantees (Theorems 8 and 9) are the first such results for the practical version of the random forest algorithm. We hope this clarifies the main points of our work. In the revised manuscript, we will clarify the goals of our work and add a complete description of the RFPI algorithm for which our theory provides coverage guarantees.
> 2. The stability we are considering is also referred to as "out-of-sample stability" in Ref. [3]. The parameters $(\varepsilon,\nu)$ characterize a "spectrum" of how stable an algorithm is (per Professor Barber's recent talk on stability of black-box algorithms), and in general there is no requirement that $(\varepsilon,\nu)$ must converge to 0 for a given algorithm. That said, a vanishing pair of $(\varepsilon,\nu)$ is practically appealing. We show in Theorem 7 that such parameters depend on the data distribution, number of data points $n$, and number of trees $B$ in the random forest algorithm. For non-heavy-tailed distributions, letting $n$ and $B$ go to infinity can result in vanishing $(\varepsilon_{n,B},\nu_{n,B})$. In the revised manuscript, we will present the "specific expression for $\nu_{n,B}$, and make necessary discussions," as suggested by the reviewer, which was partially addressed in the Appendix (C.4).
> The tightness of the convergence rate is relevant. An inspection of our proof shows that there is some freedom in choosing the dependence of $B$, as well as of $\lambda$, on $n$ to have vanishing $(\varepsilon_{n,B},\nu_{n,B})$, and the optimal rate depends on the data distribution (through $\mathbb{E}[Z_{(n)}^2]$). However, $\mathbb{E}[Z_{(n)}^2]$ originates from the bound for $|\mathsf{rf}-\mathsf{rf}^{\backslash i}|$, and it is unclear whether this is the tightest bound in general, as it only takes advantage of the range of the predicted values of tree predictors. Whether we can find a more informative bound for $|\mathsf{rf}-\mathsf{rf}^{\backslash i}|$ is an open problem, and only after this is resolved can we discuss the optimal rate of convergence for $(\varepsilon_{n,B},\nu_{n,B})$. To find such an improved bound is one of our future research directions, as stated in Sec. 3.4.
> 3. By light-tail we mean non-heavy-tail. Please note that "light tail" appears in the informal Theorem 1 and other qualitative discussions, and when formally presenting our stability result with vanishing $(\varepsilon_{n,B},\nu_{n,B})$ in Corollary 3, we technically use sub-gamma distributions, which are well defined. We understand that "light tail" could be confusing for some readers, and we will make revisions accordingly.
> 4. The reviewer mainly addresses two points here: "new technical proof" and "new algorithm." For the first point, we want to emphasize that our aim in this work is not to provide new techniques. Rather, we use recently established techniques (stability by bagging) and standard concentration inequalities to provide new understanding of the random forest algorithm used in practice. We show that the practical version of random forests is stable, and the previously proposed methods to construct prediction intervals are almost provably valid approaches, both under mild conditions. These are the two major contributions of this work. (Corollary 4 is also a new result for the jackknife-with-stability method.) We don't believe that the techniques in our proofs are "simple extensions of current techniques." There are at least two key developments: first, we notice the range of tree predictors is bounded conditioned on training data, which does not hold for a general learning algorithm; second, we take into account that $B_i$, the number of times the $i$th data point is not included in a bootstrap sample, is random, rather than fixed. We also provide the complete proof of an important result, Lemma 5, in the Appendix, and we establish the upper bound result by noticing the symmetry in the probabilistic deviation bound in the Appendix (E). None of this work is trivial. For the second point, again, it is not our aim in this work. However, our stability results do inspire new algorithms in other topics. Some of our ongoing work uses the stability property of random forests to perform active learning, which will be reported elsewhere.
> 5. We will cite the suggested papers. Please note that although there are many theoretical results for random forests, few apply to the greedy random forest used in practice. Our work applies to many variants of random forests, including the greedy one, which makes it particularly relevant in theory and practice.
>
> We hope after we fully address the comments, the reviewer will consider our work important and raise the score.

---

### Official Review · Reviewer_mxxw · 2023-07-06

**Soundness:** 4 excellent
**Presentation:** 3 good
**Contribution:** 3 good
**Rating:** 7
**Confidence:** 4

**Summary:**

Random forests are one of the most used Machine Learning methods. Its standard variant (for regression) takes the following form. GIven a random sample $D=(X_i,Y_i)_{i\leq n}$ of covariate/response pairs, one takes $B$ bootsrapped samples from $D$ and trains a tree regressor on each boostrapped sample (using one of many possible criteria). The estimated regression function $RF(x)$ is the average of the $B$ tree regressors on $x$.

Theorem 1 of the present paper is a kind of stability property for $RF$ vis a vis its "out-of-bag" variant $RF^{\backslash i}$. Here $i\in [n]$ and $RF^{\backslash i}$ is the version of $RF$ where bootstrapped samples not containing the $i$-th sample point are discarded. The paper then shows that one can use this stability property to build predictive intervals from the "OOB residuals" $|Y_i-RF^{\backslash i}(X_i)|$. Theorem 2 proves that these intervals, after a slight enlargement, provide good coverage at near-nominal-levels, whereas Theorem 3  gives exact $1-\alpha$ coverage in the limit, under suitable assumptions.

One of the main assumptions the authors impose for these last results is that the response variables be light-tailed. However, a small set of experiments suggests that similar properties would hold even for very heavy tailed covariates.

The present paper joins string of recent works dealing with uncertainty quantification for ML methods without resorting to a calibration sample (in which case one could use conformal prediction). The authors argue that main distinguishing feature of Theorems 2 and 3 is that their predictive intervals are much less demanding computationally. As computing the OOB predictors comes as a byproduct of the RF computations. By contrast, related work (eg. on jacknife type estimators) require the regression method to be rerun several times on leave-one-out samples.

Let me also say a few words about proofs. The idea is to first explore the $B\to +\infty$ limit of $RF$ and $RF^{\backslash i}$ and prove stability in that setting, via arguments from arXiv:2301.12600.

**Strengths:**

As noted above, the paper obtains a lightweight method to compute (nearly-)valid prediction intervals from random forests, which are often used in practice. This means that the result is significant (though see the next field). It also seems to be original. The exposition is fairly clear.

**Weaknesses:**

* Mathematically, it seems that much of the work behind the paper comes from arXiv:2301.12600 by Soloff, Barber and Willett.
* The nonasymptotic Theorem 2 is a bit unsatisfying in that both the interval length and the coverage are subject to errors that are hard to quantify in practice (still, since people will use RF anyway, it's nice to have some result of this kind).
* The results do not give a good bound on how large $B$ needs to be in order for the method to work well. This is clearly related not just to the maximum $Z_{(n)}$, but to how small an error one wants to allow when defining the interval.
* One very minor comment: Theorem 3.4 from arXiv:2301.12600 (quoted in line 163) should be Theorem 9 (the numbering seems to have changed from v1).

**Questions:**

1) It seems from reading arXiv:2301.12600 that $Z_{(n)}$ could be replaced with the range $\max_{1\leq i<j\leq n}|Y_i-Y_j|$ throughout the analysis. Is this true? One advantage of the range is that it is centered.

2) The authors obtain what is called a "marginal coverage" guarantee: if one single extra point $(X,Y)$ is given, then everything work. A stronger guarantee would be this: letting $\mu$ denote the distribution of $(X,Y)$

$$P\{\mu\{(x,y): |y-RF(x)|\leq  q_\alpha(R_i) + \epsilon\}\geq 1-\alpha-\nu.$$

Could this stronger guarantee be obtained via the present methods.

3) As noted in a previous field, computing how large $B$ needs to be seems to be a pressing problem. Checking the bound in Theorem 12 in arXiv:2301.12600 (v2), it seems that the present paper skips the variance-based proof for the variability of $RF_B$ and goes straight to the range-based bound. This means that the parameter $\epsilon_{n,B}$ in Theorem 2 is a sum of two parts: one that would be there even if $B=+\infty$, and a second term that comes from finite $B$. It seems to me, however, that this second term could be controlled via suitable empirical concentration inequalities such as Lemma E.4 in https://arxiv.org/pdf/2212.09900.pdf: in practice, this would mean that the second error term just mentioned could be controlled for finite B. This, in turn would make the method a bit heavier, but more quantifiable. Could the authors comment on this?

4) Regarding Theorems 3 and 10: it would seem that a sufficient condition for the theorem to hold is that
$Y-\mathbb{E}[Y\mid X]=:\eta$ satisfies:
$$\lim_{h\searrow 0}\mathbb{E}[\sup_{x,y\in\mathbb{R}\,:\,|x-y|\leq h}|F_{\eta\mid X}(x)-F_{\eta\mid X}(y)|]=0.$$
where $F_{\eta\mid X}$ is the conditional cdf of $\eta$ given $X$. This kind of condition is very natural: eg. it holds if the density of $\eta\mid X$ is uniformly bounded. Would it indeed be sufficient to require this condition on $\eta$?

**Limitations:**

There is no explicit discussion of limitations, nor do I think one would be necessary.

---

> ### Author Rebuttal · Authors · 2023-08-08
>
> We thank the reviewer for the positive comments on our work. In particular, we are happy that the reviewer considers that "the result is significant" and "original," obtaining "a lightweight method to compute (nearly-)valid prediction intervals from random forests, which are often used in practice." The most important message in our work is indeed that random forests can be used to construct justified prediction intervals at almost no extra computational cost. Below are our point-by-point responses to the reviewer's comments in "Weaknesses" (W) and "Questions" (Q).
> 1. (W1) We agree that Ref. [25] (arXiv:2301.12600) serves as an important foundation of this work. However, there are two key developments in this work to achieve the random forest stability. First, we consider regression problems where the base tree learner does not output unconditionally bounded predictive values. This is tackled by introducing the conditional stability property first and then averaging over the data distribution. As such, we arrive at the out-of-sample stability as discussed in Ref. [3], which is different from the absolute stability (Definition 4 in the updated version of Ref. [25]). Definition 11 in the updated version of Ref. [25]  is like our definition of conditional stability where a data-dependent range is involved. Second, in our work, each $B_i$, the number of times the $i$th data point is not included in a bootstrap sample, is random, while in Ref. [25], it is fixed to be a constant. Our setting is technically trickier, as the lack of a universal constant lower bound for $B_i$ makes the analysis more difficult and results in a slower convergence rate for the $|\mathsf{rf}^{\backslash i}-\mathsf{RF}^{\backslash i}|$ term.
> 2. (W2) We also agree that the stability parameters can be hard to quantify in practice. Nonetheless, the most important point of our work is to qualitatively prove that the practical version of random forests is stable under mild conditions, and we hope our work will inspire more research in this direction. We thank the reviewer for considering that "it's nice to have some result of this kind."
> 3. (W3) We thank the reviewer for this insightful question. On one hand, the current framework of analysis involves many factors: the data distribution, the dependence of $\varepsilon_i$ and $\lambda$ on $n$, etc. For such relatively simple situations as $Y$ is bounded, from Corollary 5 in the Appendix, we can conclude that $B$ does not need to be extremely large. A sublinear dependence on $n$ suffices. However, there are six terms in $\varepsilon_{n,B}$ and $\nu_{n,B}$, and there is some freedom in choosing each term's convergence rate, which may also depend on factors other than $B$, such as the dependence of $\varepsilon_1$ on $n$. Rather than look for some "optimal" convergence rate, we focus on presenting the important qualitative result that $\varepsilon_{n,B}$ and $\nu_{n,B}$ can converge to 0 under mild conditions. On the other hand, the current framework of analysis itself is not necessarily the "optimal" one. For example, we have required $\varepsilon_1=\varepsilon_3$ in our proof for simplicity, but doing so removes a degree of freedom. Moreover, all the analysis in this work is built on the upper bound $2Z_{(n)}$ for the difference between $\mathsf{rf}$ and $\mathsf{rf}^{\backslash i}$. This bound could be crude for a typical dataset, as hinted by our numerical results in Sec. 3.4. So, at this stage, we focus on the qualitative aspect of our work, leaving the quantitative improvements as future study.
> 4. (W4) Thanks for pointing out this change in the updated version of Ref. [25]. We will revise our manuscript accordingly.
> 1. (Q1) Yes. $\max_{i<j}|Y_i-Y_j| \leq 2Z_{(n)}$, both serving as a measure of the spread of $Y$. We think your statement that "One advantage of the range is that it is centered" refers to that $Y_i-Y_j$ is centered. Please let us know if this is not what you mean. The main reason we use $Z_{(n)}$ is that in some cases, the scaling or even the exact expression of $\mathbb{E}[Z_{(n)}^2]$ is handy to use, thus simplifying the analysis.
> 2. (Q3)  We are not completely sure about the LaTeX expression, which seems to address the coverage probability conditioned on the training set. If so, then this is a very insightful question with strong practical relevance. Our theory basically follows the idea of conformal prediction, and thus cannot provide guarantees conditional on the training set. We note there are several recent works that try to generalize the marginal coverage to various kinds of conditional coverages such as arXiv:2305.12616. We hope in the future some conditional coverage of random forest prediction intervals can be established.
> 3. (Q3) Thanks very much for raising the point of using a potentially sharper concentration inequality in our analysis. We believe the situation is like the choice between variance tensorization (Theorem 2.3 in Ramon van Handel's notes "Probability in High Dimension") and bounded difference inequalities (Corollary 2.4, ibid.). Our choice of using $2Z_{(n)}$ as the bound corresponds to the latter, which is technically more tractable when a precise calculation of the variance is not easy. If there is a way to calculate the variance, of course a better bound can be established. (But at this moment, we are unaware of any such results.)
> 4. (Q4) The asymptotic coverage that we establish concerns the continuity property of the CDF of the residual $|Y-\mathsf{RF}(X)|$, and there is no requirement for the CDF of $Y-\mathbb{E}[Y|X]$. As $Y-\mathsf{RF}(X)$ does not necessarily converge to $Y-\mathbb{E}[Y|X]$ in probability, the CDF of the latter is not relevant.
>
> We greatly thank the reviewer for a careful reading of the technical details, useful suggestions, and many insightful questions.

---

> > ### Comment · Reviewer_mxxw · 2023-08-14
> > **Thank you**
> >
> > Thank you very much for the rebuttal. I have nothing else to add at this point.

---

### Official Review · Reviewer_sZkt · 2023-07-06

**Soundness:** 3 good
**Presentation:** 3 good
**Contribution:** 3 good
**Rating:** 6
**Confidence:** 4

**Summary:**

In this paper the authors considers the issue of stability of the often used in practice Random Forest algorithm and provide theoretical bounds on the $\varepsilon$-stability upto an order of $O_{\mathbb{P}}(|Y|^2_{(n)}/n)$ (i.e. the largest in magnitude observation) when fitting the method with $n$-iid sample points $(X_i,Y_i)$. A light tailed assumption on $Y^2$ thereby yields suitable control over the asymptotic behavior of $|Y|^2_{(n)}/n$. Further, the stability results are used to derive $n$-dependent lower and upper bounds (under increasingly more assumptions) for the coverage probability of prediction intervals constructed from out-of-bag error of random forests. In comparison to many other results in the literature, this paper works with a practical version of random forests.

**Strengths:**

1. In comparison to many other results in the literature, this paper works with a more practical version of random forests.

**Weaknesses:**

1. It was not clear whether the stability bounds were optimal or could be improved.
2. The presentation usually benefits from working under a single set of assumptions instead of increasingly more assumptions (however this is a minor point).


**Questions:**

If one assumes that $Y=f(X)+\eta$ for $\eta \sim F$ for some suitable light tailed distribution and $f\in \mathcal{F}$ in some classical function class, can one have modified rates for stability bounds depending on $F$ and $\mathcal{F}$?

**Limitations:**

None noted

---

> ### Author Rebuttal · Authors · 2023-08-07
>
> We thank the reviewer for considering that "this paper works with a more practical version of random forests." We also believe the most important point of our results is that they apply to the practical version of random forests, and are thus strongly relevant to applied machine learning. We theoretically prove in this work that random forests can be used to provide justified prediction intervals at almost no extra computational cost, which is appealing particularly for tasks where uncertainty quantification is desired and the computing resource is limited. Below are point-by-point responses to the reviewer's "Weaknesses" and "Questions" comments.
> 1. "It was not clear whether the stability bounds were optimal or could be improved." We agree with the reviewer on this point. On one hand, our bound is valid for a training set that contains at least two data points, and it seems that there is not much room for further improvement in the most general situation. On the other hand, for a dataset of typical size in practice, and for non-pathological data generating distributions, our numerical experiment (Sec. 3.4 of the manuscript) seems to suggest that the current bound could be improved. One of our research goals is to find a more informative bound that holds at least for TYPICAL cases, while the current bound we use holds for the WORST case. By providing the first stability result for random forests, we hope our work can draw the attention of more researchers to work towards a more satisfactory bound.
> 2. "The presentation usually benefits from working under a single set of assumptions instead of increasingly more assumptions (however this is a minor point)." We agree that stating all assumptions first makes the presentation of results clearer. Our structure basically follows previous works such as Ref. [17]. We start with the stability result (Theorem 7) and the coverage lower bound (Theorem 8), which hold under minimal assumptions and already carry sufficient useful information to guide machine learning practice. In many works on distribution-free  prediction, such as Refs. [3], [15], the aim is to establish the lower bound. Statisticians such as the authors of Ref. [17] also pay much attention to the upper bound, which cannot be obtained without more assumptions. Such assumptions are mild, and can usually be satisfied by typical data distributions. Hence we also include this in our work (Theorem 9). Lastly, in order to compare with a previous asymptotic result on the coverage of random-forest prediction intervals in Ref. [32], we also derive a similar one (Theorem 10). This part is somewhat independent from the stability result and non-asymptotic lower and upper bounds. The assumptions made in Theorem 10 are also less intuitive than those made in Theorems 7-9. To highlight the most important contributions, we decided to state the assumptions in the present way. We thank the reviewer and hope our structure can be considered as reasonable.
> 3. If the true model is known, "can one have modified rates for stability bounds?" Thanks for raising this insightful question. While our theory provides a sufficient condition for stability of random forests, we do not expect its optimality in all cases. If we do know the true model, then we might perform some other analysis by fully taking advantage of the information about the data distribution and the function class. As long as one is able to come up with a more informative bound for $|\mathsf{rf}-\mathsf{rf}^{\backslash i}|$, the convergence rate can be modified. In the present work, our focus is the model-agnostic case. Again, our work serves as an important initiation point towards research in random forest stability, and we leave the quantitative improvement of the present work as future study.
>
> We hope we have fully addressed the reviewer's comments, and have made clearer the qualitative importance and practical relevance of our work.

---

### Official Review · Reviewer_1UHc · 2023-07-07

**Soundness:** 4 excellent
**Presentation:** 4 excellent
**Contribution:** 3 good
**Rating:** 8
**Confidence:** 3

**Summary:**

The paper presents new and strong set of results on stability of (greedy version of) random forests. Theoretical (resp. numerical) evidence is provided to support stability for light-tail (and heavy-tail) assumptions on marginal distribution of squared response. New finite sample upper and lower bounds are provided for prediction intervals constructed from OOB effort of random forests. The paper can be regarded as a demonstrative work that justifies the merit of random forests for both point and interval prediction.





**Strengths:**

The style of results and overall content of paper are very appealing. I like the way in which stability and prediction interval results were informally stated first and then rigorously discussed later. A clear review of algorithmic stability concepts helped me understand the proof ideas.



**Weaknesses:**

-- I think the writing can be slightly improved by articulating the use of absolute stability results for bagged algorithms.
-- The transition from derandomized version of RF to finite-B case through the route of conditional stability analysis is a bit abrupt and needs more explanation, particularly with regard to concentration of measure.
-- The limitation/extension of theory to heavy-tailed case is interesting but some comments about the key bottlenecks in the proof technique that would need to be overcome to achieve such a generalization would make the contribution stronger. Also where does the theory break? The experimental results are not very clear in this regard.
-- On the prediction interval part, Table 1 can be enhanced and repositioned to include computational advantages as well. Finally, the discussion around “jackknife-after bootstrap-with-stability (JaBS)” is a bit hard to follow since the discussion on J+aB and J+aBS is intertwined with the intuition on how subsequent results in the paper build on stability results – it is desirable to streamline this discussion a bit.
-- please consider citing and discussing relevant work in OR on optimal classification/prescriptive trees by Bertsimas and co-authors as well as their work on Stable Classification. Your approach and focus is different, but given the stated goal of putting the stability of random forests on a stronger footing, it may be worthwhile to make a connection.

**Questions:**

Please see my suggestions above.

**Limitations:**

Everything on this front seems reasonable.

---

> ### Author Rebuttal · Authors · 2023-08-07
>
> We gratefully thank the reviewer for the positive comments of our work, and the suggestion of a “Strong Accept.” The reviewer states that “the paper can be regarded as a demonstrative work that justifies the merit of random forests for both point and interval prediction.” This statement contains exactly the message we want to convey to the machine learning community: for typical tabular datasets, random forests can provide justified interval prediction at almost no extra computational cost, in addition to good point prediction. We believe our theoretical results in this work can have a strong impact on machine learning practice in various fields. In the revised manuscript, we will fully address the points raised by the reviewer to further improve our work. Below are our point-by-point responses to the reviewer’s “Weaknesses” comments.
> 1. “Articulating the use of absolute stability results for bagged algorithms.” This kind of stability is addressed in Ref. [25] and serves as an important foundation of our work. In this case, stability can be established with the only requirement that the base learner outputs bounded predicted values, and the data distribution is irrelevant. This can be appealing in certain tasks such as 2-class classification, where $Y$ itself is naturally bounded. One can use logistic regression, the tree predictor, or any other learning algorithm with bounded output as the base learner, and by aggregating results obtained on bootstrapped samples, a stable predictor can be achieved, in the sense that arbitrarily removing a single training data point will not much influence its prediction on a future data point. (It is worth stressing that in general regression tasks, $Y$ is unbounded, so requiring base learners to output unconditionally bounded predictive values can be a problem. However, inspired by the underlying idea of absolute stability, we in this work address this problem by first considering conditional stability and then averaging over the data distribution.)
> 2. "The transition from derandomized version of RF to finite-B case...needs more explanation." The derandomized version can be effectively seen as the $B=\infty$ case. That is, each bootstrap sample is drawn with equal probability an infinite number of times, resulting in the population mean $\mathsf{rf}(x)$ of tree predictors for any future $x$. In practice, $B$ is finite, and each bootstrap sample is drawn with equal probability a finite number of times, resulting in the sample mean $\mathsf{RF}(x)$ of tree predictors. The difference between $\mathsf{rf}(x)$ and $\mathsf{RF}(x)$, when conditioned on training data, can be quantified by Hoeffding's inequality, a standard concentration inequality. This approach was also used in previous works, c.f. Refs. [15], [25].
> 3. The issue of heavy-tail distributions and bottlenecks of our theory. Our proof mainly relies on the upper bound $2Z_{(n)}$ of the difference between $\mathsf{rf}$ and $\mathsf{rf}^{\backslash i}$ when conditioned on training data. It is a valid bound for many variants of random forests and for any training set with size no less than two. However, it is unclear whether this bound is tight in general. Developing our results on a possibly crude bound leads to the conclusion that the random forest is stable when $Y$ is not heavy tailed. Please note that this is a SUFFICIENT, rather than necessary, condition for random forest stability. Numerically, we found that even for heavy-tailed distributions, the stability seemed to persist. This indicates that there might exist a more informative bound, at least for non-pathological data generating distributions and for datasets of typical size. Once such an informative bound is established, our theory can be modified to more thoroughly explain the experimental result. Our work serves as an important initiation point for research in this direction.
> 4. "Table 1 can be enhanced and repositioned to include computational advantages." We will revise the manuscript accordingly. Also, with an extra one page for accepted papers, we expect to be able to provide the information of guaranteed coverages of those algorithms as well.
> 5. The discussion around JaBS is hard to follow. We are sorry for the unclarity. The length limit forced us to write it this way. We will streamline this discussion in the revised manuscript as suggested.
> 6. Bertsimas and co-authors's work. Thanks for introducing this line of work to us. It is not only relevant, but also inspiring. For example, combining the JMLR paper "Stable Classification" and the "stability by bagging" idea seems to immediately suggest the method of "importance aggregating." That is, we aggregate the best-performing base learners with more weights, which are trained on some "good" bootstrap samples. This can be a potential topic. We will discuss and cite the relevant papers.
>
> We hope we have fully addressed the points raised by the reviewer, and we appreciate the reviewer's valuable suggestions.

---

> > ### Comment · Reviewer_1UHc · 2023-08-14
> >
> > Thanks for responding to my review. No more clarifications sought from my side.

---

### Author Rebuttal · Authors · 2023-08-08

We want to thank all reviewers for their time and comments.

We are encouraged that three of them give positive evaluations to our work and suggest to accept our manuscript with ratings 6, 7, and 8, respectively. All three reviewers agree that our theoretical work is relevant for applied machine learning. Actually, this is the most important message we want to convey in this work. We prove that, under mild conditions, the practical version of random forests can be used to construct justified prediction intervals at almost no extra computational cost. Given the well-known fact that random forests have good performance on point prediction, our result "can be regarded as a demonstrative work that justifies the merit of random forests for both point and interval prediction," as per Reviewer 1UHc. We appreciate this reviewer's summary of our work, and we are happy to share it with more machine learning researchers in various fields. Following Reviewer BF9X’s suggestions, we will further clarify our problem, conclusions, the definitions of some terms, and the novelty and significance. We will try our best to make the revised manuscript more accessible to a broader readership.

Each reviewer's comments are addressed separately below.

---

### Decision · Program_Chairs · 2023-09-21

**Decision:**

Accept (poster)

**Comment:**

Three out of four reviewers have acknowledged the strength of the contributions of the paper. Even though some concerns have been raised at the initial reviews, the authors have addressed them adequately. Hence, I am recommending an acceptance. Please implement all the changes promised in the rebuttal.